# A Simple and Efficient Method for Preparing High-Purity α-CaSO_4_·0.5H_2_O Whiskers with Phosphogypsum

**DOI:** 10.3390/ma15114028

**Published:** 2022-06-06

**Authors:** Yan Lin, Hongjuan Sun, Tongjiang Peng, Wenjin Ding, Xiang Li, Sha Xiao

**Affiliations:** 1Key Laboratory of Solid Waste Treatment and Resource Recycle, Ministry of Education, Mianyang 621010, China; linyan3345@mtc.edu.cn (Y.L.); tjpeng@swust.edu.cn (T.P.); dingwenjin@swust.edu.cn (W.D.); lx17861511331@163.com (X.L.); 13541207191@163.com (S.X.); 2Institute of Mineral Materials and Application, Southwest University of Science and Technology, Mianyang 621010, China; 3Analytical and Testing Center, Southwest University of Science and Technology, Mianyang 621010, China

**Keywords:** phosphogypsum (PG), CaSO_4_·0.5H_2_O (HH) whiskers, CaSO_4_·2H_2_O (DH) whiskers, cycle number, sustainable development

## Abstract

A simple and efficient approach for the high-purity CaSO_4_·2H_2_O (DH) whiskers and α-CaSO_4_·0.5H_2_O (α-HH) whiskers derived from such phosphogypsum (PG) was proposed. The impact of different experimental parameters on supersaturated dissolution–recrystallization and preparation processes of α-CaSO_4_·0.5H_2_O was elaborated. At 3.5 mol/L HCl concentration, the dissolution temperature and time were 90 °C and 20 min, respectively. After eight cycles and 5–8 times cycles, total crystallization amount of CaSO_4_·2H_2_O was 21.75 and 9.97 g/100 mL, respectively, from supersaturated HCl solution. The number of cycles affected the shape and amount of the crystal. Higher HCl concentration facilitated CaSO_4_·2H_2_O dissolution and created a much higher supersaturation, which acted as a larger driving force for phase transformation of CaSO_4_·2H_2_O to α-CaSO_4_·0.5H_2_O. The HCl solution system’s optimum experimental conditions for HH whiskers preparation involved acid leaching of CaSO_4_·2H_2_O sample, with HCl concentration 6.0 mol/L, reaction temperature 80 °C, and reaction time 30 min–60 min. Under the third cycle conditions, α-CaSO_4_·0.5H_2_O whiskers were uniform in size, clear, and distinct in edges and angles. The length range of α-CaSO_4_·0.5H_2_O whiskers was from 106 μm to 231 μm and diameter range from 0.43 μm to 1.35 μm, while the longest diameter ratio was 231. Purity of α-CaSO_4_·0.5H_2_O whiskers was approximately 100%, where whiteness reached 98.6%. The reuse of the solution enables the process to discharge no waste liquid. It provides a new reference direction for green production technology of phosphogypsum.

## 1. Introduction

Phosphogypsum (PG) is mainly composed of CaSO_4_·2H_2_O (DH), which accounts for approximately 90% of its components and is a waste residue produced by the limestone-gypsum wet-process using phosphoric acid technology [1,2]. PG waste is produced at a rate of 5 tons of PG per ton of phosphoric acid and amounts to approximately 200 million tons per year worldwide [3,4,5]. PG contains impurities such as water-soluble phosphate (P_2_O_5_), fluorine (F), organic matter, heavy metals, and radioactive elements [6,7]. Currently, most PG is stockpiled outdoors, where the bulk deposition of PG can result in not only large land occupation but can also cause environmental pollution in soil, water, and air [8,9,10]. Hence, the wet phosphoric acid (WPA) process industry is currently considered a non-green industry and is the reason why PG has drawn much attention in recent years. In the U.S., PG is a huge problem since it is a big liability for the taxpayer, and Europe seems to be looking at ways to use PG as a gypsum replacement [11]. In China, the government has decided to accelerate the promotion of increasing phosphogypsum inventory to zero in accordance with the principle of “whoever discharges the slag will take care of it, and whoever uses it will benefit”. Thus, the disposal of phosphogypsum is an urgent matter for taxpayers whether in China, the United States, or other countries.

PG waste is widely used as an alternative to natural gypsum owing to its similar chemical composition and binding feature and is a primary resolution to the green development of the phosphoric acid industry. At present, PG is extensively applied in fields such as building materials [12,13,14], road materials [15,16,17], agriculture [18,19,20], and other applications [21,22,23]. However, these applications have relatively low additional value, so it is crucial to improve the additional value of phosphogypsum.

Furthermore, CaSO_4_·0.5H_2_O (HH) whiskers are a new type of inorganic fiber material with high added value, which is advantageous in the filler market due to its superior workability, high strength, and favorable biocompatibility [24,25]. Nowadays, the industrial production process of α-CaSO_4_·0.5H_2_O (α-HH) whiskers mainly includes an autoclave method and pressurized aqueous method. Both processes require high temperature and pressure, which consumes more energy. In contrast, the non-autoclave method of salt or acid solution has been increasingly applied because of its mild reaction conditions and reduced energy consumption [25].

Herein, DH is extracted in order to purify PG due to supersaturated crystallization, followed by preparation of α-HH whiskers by the non-autoclave method of HCl acid solution with purified PG as raw material. The studied method provides a simple way to prepare α-CaSO_4_·0.5H_2_O whiskers that can meet the industrial requirements of the double 98 standard (whiteness and purity) using phosphogypsum. Additionally, factors affecting the whiskers are studied and discussed, providing a new reference for high additional value utilization of PG.

## 2. Experimental Section

### 2.1. Materials

The chemical composition of phosphogypsum (PG) is given in Table 1. PG was obtained from Chinese waste residue produced by limestone-gypsum wet-process phosphoric acid. The radioactive test results of phosphogypsum are given in Table 2. The phosphogypsum sample conforms to Class A material, according to the NATIONAL standard GB6656-2010. Ethanol and hydrochloric acid were purchased from Kelong Chemical Reagent Co., Ltd., Chengdu, China.

### 2.2. Methods

Acid leaching was performed by subjecting the PG to HCl solution. Ten grams of PG was placed into a 125 mL reaction vessel, to which 100 mL of HCl solution was added. A magnetic stir bar was used to ensure adequate mixing throughout the reaction. A hot bath was used to maintain the desired reaction temperature. After the desired reaction time, the mixture was cooled and filtered to separate the solids from the solution. The optimum conditions for dissolution and extraction, including hydrochloric acid concentration (N), temperature (T), time (S), and cycling conditions (X), are listed in Table 3. CaSO_4_·2H_2_O, extracted after 4–8 times cycles and used as raw material, was named PYJ. This was poured into higher concentration of HCl solution and stirred with a water-bathing constant temperature vibrator at a volatile rate of 200 r/min. After desired reaction temperature and time, the solution was filtrated to produce α-CaSO_4_·0.5H_2_O whiskers, which were washed thrice with distilled water and rinsed once with ethanol before being dried at 60 °C for 12 h in an oven. Factors and levels of α-CaSO_4_·0.5H_2_O whiskers production experiments are listed in Table 4.

### 2.3. Data Analysis

The phase composition of the produce was studied using X-ray diffractometer (XRD, Ultima IV, Akishima, Japan) on randomly oriented specimens. The semi-quantitative evaluation of phases was performed based on the integrated intensity of the main peak of the single components, considering the respective reference intensity ratios (I/Ic or RIR) with respect to a reference phase (corundum). The concentration of phase A, for example, was calculated according to the following relation [26]:A = 100 × (I_A_/RIR_A_)/(I_A_/RIR_A_ + I_B_/RIR_B_ + I_C_/RIR_C_ + … + I_N_/RIR_N_)(1)
where A + B + C + … + N = 100, I_A_, I_B_, I_C_ … I_N_ were the integrated intensities of the main peak of phases A, B, C … N. The RIR values of DH, HH, and AH were referenced from the PDF 76–1746, 81–1848, and 72–0503, respectively. The measurements were conducted using scanning speed of 0.02 deg/s, X-ray tube voltage of 40 kV, and tube current of 40 mA. Scanning electron microscopy (SEM) of the samples was performed using Zeiss Ultra 55 microanalyzer with an accelerating voltage of 15 kV. The sample whiteness test was conducted with SBDY-1P, and reference whiteboard was R457:84. We used CIT-3000F building materials radioactive detection analyzer to test phosphogypsum sample, according to the NATIONAL standard GB6656-2010. Differential thermal analysis and thermal–gravimetric analysis (TG-DTA) of products was analyzed using thermal analyzer method with SDTQ6000 analyzer using air atmosphere heating rate of 10 °C/min. The dissolution amount of PG and crystallization amount of CaSO_4_·2H_2_O were weighed with electronic weighing instrument. Extraction rate of CaSO_4_·2H_2_O was the percentage of the dissolution amount of PG to the crystallization amount of CaSO_4_·2H_2_O.

### 2.4. Theoretical Basis

Gypsum may crystallize in three modifications, namely dihydrate (DH), hemihydrate (HH), and anhydrite (AH). The chemical formula of gypsum is CaSO_4_·*n*H_2_O (*n* = 0, 0.5, 2). The dissolution of CaSO_4_·*n*H_2_O in aqueous solution is expressed by Equation (2):
CaSO_4_·*n*H_2_O = Ca^2+^ + SO_4_^2−^ + *n*H_2_O (*n* = 0, 0.5, 2)(2)


The equilibrium constants for Equation (3), that is, solubility product constants [27,28]. The product of its concentration c and the corresponding activity coefficient γ determines the activity of an ion.

(3)$$K_{SP}=[{\mathrm{Ca}}^{2+}]eq\times [{\mathrm{SO}}_{4}^{2-}]eq\times {{[H}_{2}O]}_{eq}^{n}\;(n=0,\;0.5,\;2)$$(4)α=c×γ(5)$$I=\frac{1}{2}\sum c_{i}Z_{i_{}}^{2}$$
where, in the dilute solution, c represents the concentration of the substance, γ corresponds to the ionic activity coefficient, and *I* corresponds to ionic strength. $c_{i}$ and $Z_{i}$ are ionic concentration and ion charge number, respectively [29]. Therefore, the ionic strength *I* increases when the ionic concentration c increases.
(6)$$-\mathrm{lg}\gamma_{i}=AZ_{i}^{2}\frac{\sqrt{I}}{1+B\overset{0}{a}\sqrt{I}}$$
where the ionic activity coefficient is calculated according to Debye–Hückel’s formula in dilute solution. A and B are functions of the thermodynamic temperature T and the solvent dielectric constant, respectively. $\overset{0}{a}$ is ionic volume parameter [30]. Therefore, the activity coefficient ions γ decrease when the ionic strength increases. It is therefore easy to explain the increase of the gypsum’s solubility product with the introduction of HCl in the solution.
(7)$$S=\frac{a{\mathrm{Ca}}^{2+}\cdot a{\mathrm{SO}}_{4}^{2-}\cdot {aH_{2}O}_{}^{n}}{K_{SP}}\;\;\;(n=0,\;0.5,\;2)$$
where *S* is the saturation ratio, α is the activity of the solute, and *K_SP_* is the solubility product of gypsum [31]. Acid effect can promote the dissolutio $\gamma_{i}$
*n* of PG and phase transformation. The whole preparation process involves the following essential chemical reactions (8)–(13).

Extracting process:PG (s) + HCl (aq) → Ca^2+^ (aq) + SO_4_^2−^ (aq) + H^+^ (aq) + Cl^−^ (aq) + SiO_2_ (s)(8)
Ca^2+^ (aq) + SO_4_^2−^ (aq) + 2H_2_O → CaSO_4_·2H_2_O (s)(9)

Preparation process of α-HH:CaSO_4_·2H_2_O (s) + HCl (aq) → Ca^2+^ (aq) + SO_4_^2−^ (aq) + H^+^ (aq) + Cl^−^ (aq) + 2H_2_O(10)
Ca^2+^ (aq) + SO_4_^2−^ (aq) + 0.5 H_2_O → α-CaSO_4_·0.5H_2_O (s)(11)
α-CaSO_4_·0.5H_2_O (s) + HCl (aq) → Ca^2+^ (aq) + SO_4_^2−^ (aq) + H^+^ (aq) + Cl^−^ (aq) + 0.5 H_2_O(12)
Ca^2+^ (aq) + SO_4_^2−^ (aq) → CaSO_4_ (s)(13)

## 3. Results and Discussion

### 3.1. CaSO_4_·2H_2_O Extraction from PG

#### 3.1.1. Effect of the Dissolution Amount, Crystallization Amount and Extraction Rate

Dissolution recrystallization of gypsum is the key process for DH extraction from PG. Single-factor experiments were systematically conducted to investigate dissolution and extraction of DH in HCl solution from PG under various reaction conditions. Figure 1a shows the dissolution amount of PG and crystallization amount and extraction rate of DH in different concentrations of HCl. The results indicate that the amount of dissolved PG is comparable in 2.5 to 5 mol/L HCl, and the amount of dissolved PG increases in 1 to 2.5 mol/L HCl concentration. With the increase of concentrations of HCl, the ionic strength I is increased. As the ionic strength I of a solution increases, the activity coefficient of an ion decreases according to the Debye–Hückel limiting law [32]. Concentration c and the corresponding activity coefficient determines the activity of an ion. However, as the concentration continues to increase, temperature becomes the dominant factor, and it is not easily dissolved. However, the crystallizing amount of DH gradually increases and then decreases with increasing HCl concentration. At about 25°, increasing the concentration of hydrochloric acid can promote the growth of DH and increase the amount of crystallization. However, a too-high concentration of hydrochloric acid increases ionic strength I, which may lead to the increase of DH solubility. It is inferred that this is the reason for the slight decrease in crystal quality. Therefore, the optimum concentration of HCl is recommended when the extraction rate is maximum: the optimal concentration is 3.5 mol/L. In 3.5 mol/L HCl, the dissolution amount of sample PG is 4.99 g/100 mL, and the maximum crystallizing amount and maximum extraction rate of DH are 2.32 g/100 mL and 46.54%, respectively. The crystallizing amount of DH reaches the maximum in 3.5 mol/L HCl, and the whisker morphology is good at this concentration, so it is selected as the optimal concentration for preparing calcium sulfate dihydrate whiskers. At a given temperature, the amount of dissolved PG remains unchanged with gradually increasing HCl concentration. $K_{SP}({\mathrm{CaSO}}_{4}\cdot 2H_{2}O)$ is strongly influenced by temperature. At normal temperature, an obvious influence is observed with a dissolution amount of PG between 3 and 5 mol/L HCl. Therefore, it is possible that dissolved impurities can affect calcium sulfate crystallization [33].

Figure 1b shows the dissolution of PG and crystallization amount and extraction rate of DH at different dissolution temperatures. The results show that the dissolution amount of PG increases, and the impurity of PG easily leaches with increasing temperature. These data are in agreement with earlier studies on gypsum dissolution [34,35]. $K_{SP}({\mathrm{CaSO}}_{4}\cdot 2H_{2}O)$ is strongly influenced by temperature. Therefore, the product $a{\mathrm{Ca}}^{2+}\cdot a{\mathrm{SO}}_{4}^{2-}\cdot aH_{2}O^{2}$ increases with increasing temperature. As the temperature rises, equilibrium favors a shift in the direction of heat absorption. The temperature increase is beneficial to overcome the interaction between the substance molecules so that the substance molecules leave the surface of the substance into the solvent process. Therefore, the increase of temperature is conducive to the dissolution of DH and soluble impurities. Lowering the temperature is a common method so that S is greater than 1. When S is greater than 1, the solution naturally precipitates crystals due to supersaturated crystallization principle [31]. Firstly, the crystallizing amount and extraction rate of DH gradually increases and then reduces with increasing temperature from 60 °C to 100 °C. At 90 °C reaction temperature, the maximum crystallizing amount of DH is 2.57 g/100 mL. During the crystallization process of DH, excessive temperature hinders nucleation, and impurities in the crystal lattice of calcium sulfate are released. Hence, it is easy to explain the increase of the gypsum’s solubility product with increasing temperature.

Figure 1c shows the dissolution amount of PG and crystallization amount and extraction rate of CaSO_4_·2H_2_O at different dissolution times. The results show that they are all increased and then remain unchanged with increasing dissolve times. At 10 min dissolution time, the amount crystallized is 3.12 g/100 mL, and the extraction rate is 50.11%. When supersaturation is reached under these conditions, DH of PG will no longer dissolve. When reaction temperature returns to room temperature (25 °C ± 2 °C), it is considered that the solution changes from super-saturation to saturation, which makes it easy to explain the crystallizing amount and extraction rate of DH variation.

Figure 1d shows that the dissolution amount of PG and crystallization amount and extraction rate of H after different dissolution cycle numbers. First, the dissolution amount of PG increases and then plateaus. The cycles complete with a final, slight increase. When the number of dissolution cycles reaches between five and eight, the total crystallized amount of DH is 9.97 g/100 mL. The accumulation of PG impurities increase with the increasing number of dissolution cycles. During the crystallization process of DH, impurities in the higher impurity solution stick to the crystal nucleus and affect the crystallization amount. When the impurity reaches a certain concentration, impurities concentrate here easily, and “crystal druse” defects appear on the sites. In order to reuse the hydrochloric acid solution, only the industrial by-product gypsum can be separated from the impurities; the acid can be reused several times, and good results are obtained even after eight cycles.

#### 3.1.2. Effect of Acid-Leaching Products at Different Dissolution Cycle Numbers

Figure 2a shows XRD pattern of the crystallization product of acid-leaching solution at different dissolution cycles. At dissolution cycle times between one and eight cycles, only one phase peak of gypsum is detected in the product of acid-leaching solution. The phase peak and peak strength of the acid-leaching product of PG do not show an obvious change with increasing dissolution cycles. Hence, the number of dissolution cycles has no obvious influence on the phase and crystallinity of the acid-leaching product of industrial by-product of gypsum.

Figure 2b–i show SEM pattern of the crystallization product of acid-leaching solution at different dissolution cycles. No obvious changes are observed in the micromorphology of the crystallization product in acid-leaching solution with increasing dissolution cycles. When dissolution cycle times are less than four, the microstructure of the crystallization product are fibrous whiskers, uniform in size, with a smooth surface level. As the number of dissolution cycles (1–4) increases, the crystallization product’s maximum length–diameter ratio increases to 70, 86, 92, and 85 and the particle diameter to 2.5–30.1 μm, 2.1–13.8 μm, 2.4–18.8 μm, and 2.5–40.2 μm, respectively. At 5–8 dissolution cycles, the micromorphology of the crystallization products, which are non-uniform in size, from the leaching solution shows bunched whiskers and blocky structure. Due to the presence of fracture marks, pores, and enrichment of multi-flocculent substances on the surface, the products are not smooth. A certain amount of impurity ions in the solution is conducive to the perfection of crystal morphology, but it is obvious that the presence of more impurity ions in the solution is not conducive to the formation of whiskers. Due to the accumulation of impurity ions, it is easy to include the mother liquor and make the crystals gather together, thus forming beam crystals and irregular block crystals easily. Excessive impurities can attach to the crystal and are carried out from the supersaturated solution; it is thus easy to explain the increase in crystallization amount and extraction rate of DH after four dissolution cycles. In order to achieve zero discharge of filtrate, we used unrestricted filtrate. The author used PYJ in this paper, a mixed crystallization product of acid-leaching solution after 5–8 and even more dissolution cycles, as the raw material for the preparation of α-HH.

### 3.2. Preparation of HH Whiskers

#### 3.2.1. Effect of HCl Concentrations and Reaction Temperatures

Figure 3a shows XRD pattern of the acid-leaching products of PYJ in different HCl concentrations, in which HCl concentration has a significant influence on the phase of acid leach of PYJ. The peak intensity of gypsum decreases at first and then disappears. In the phase of HH, peak intensity increases first, and then, it completely and suddenly disappears with increasing HCl concentration, whereas that of the AH phase trends from zero and gradually increases with increasing HCl concentration until the HCl concentration reaches 6.0 mol/L only in the AH phase. According to Equation (1), DH, HH, and AH of the acid-leaching products of sample PYJ are 73.1%, 10.3%, and 16.7% in 5.0 mol/L, respectively. The DH, HH, and AH are 13.5%, 44.4% and 42.1% in 5.5 mol/L. In order to prepare high-purity HH whiskers and explore the formation process, HCl concentration of 6 mol/L is selected to discuss the influence of temperature.

Figure 3b shows XRD pattern of the acid-leaching products of PYJ at different reaction temperatures. The results show that solution temperature has a significant influence on the phase of acid leach of PYJ depending on different solution temperatures. The peak intensity of DH decreases first and then disappears, and the peak intensity of HH increases first and finally disappears, whereas that of the AH phase trends from zero and gradually increases with temperature. When reaction temperature reaches 70 °C, the AH phase peak begins to appear until reaction temperature reaches 80 °C only in the AH phase. According to Equation (1), the HH of the acid-leaching products of sample PYJ is 46.0% and 54.4% at 60 °C and 70 °C, respectively. That of DH is 54.0% and 1.7% at 60 °C and 70 °C, respectively. That of AH is 43.9% at 70 °C. In order to prepare high-purity HH whiskers and explore the formation process, reaction temperature of 80 °C is selected to discuss the influence of reaction time.

#### 3.2.2. Effect of Reaction Time

Figure 4a shows XRD pattern of the acid-leaching products of sample PYJ at different reaction times. The results show that reaction time has a significant influence on the phase of acid leach of PYJ depending on different reaction times. The peak intensity of DH decreases first and then disappears, and the peak intensity of HH increases first and finally disappears, whereas that of the AH phase trends from zero and gradually increases with the increase of reaction time. Under the same external conditions, the phase-transition processes between DH–HH and HH–AH in aqueous solutions are determined by the water activity, according to $a{\mathrm{Ca}}^{2+}\cdot a{\mathrm{SO}}_{4}^{2-}\cdot {aH_{2}O}^{n}$. When the reaction time is 15 min, the phase peak of HH begins to appear until the reaction time reaches 30 min only for the HH phase peak. When the time continues to increase to 90 min, the AH phase peak appears until 120 min only for the AH phase peak. According to Equation (1), when the reaction time is 15 min and 20 min, the DH of the acid-leaching products of sample PYJ is 35.2% and 0.2%, respectively. That of HH is 64.8% and 99.8% at 15 min and 20 min, respectively. When the reaction time range is 30 min–60 min, there is only the HH phase. When the reaction time is 90 min, the HH and AH of the acid-leaching products of sample PYJ are 60.8% and 39.2%, respectively. When the reaction time range is 120 min–180 min, there is only the AH phase.

Figure 4b–e shows SEM photos of the acid-leaching products of PYJ at 15–60 min. As shown in Figure 4b, when the reaction time is 15 min, the main morphology of the acid-leaching products are HH crystal nuclei and HH whiskers and a few clumps of DH that were not completely dissolved. Figure 4c shows that most of the morphology of the acid-leaching products is the fibrous whiskers of HH, and a small amount of the morphology of the acid-leaching products is crystal nuclei of HH at 20 min. When the reaction time is 30 min, it can be seen that the morphology of the fibrous whiskers arise almost uniform in size, with a smooth surface, clear edges and corners, a maximum aspect ratio of 136, length range from 38 μm to 180 μm, and diameter range from 0.82 μm to 1.41 μm, shown in Figure 4d. Figure 4e shows that the micromorphology of acid-leaching products are fibrous whiskers with a few broken whiskers attached on the surface, which is neither smooth nor flat after 60 min. The maximum aspect ratio is 113, with length range of 65–262 μm and diameter range of 2.33–7.1 μm. It is reasonable to hypothesize that between the dissolution times of 30–60 min, the whisker diameter increases with time, but the whisker length is too long and easily broken.

#### 3.2.3. Effect of Reaction Cycle Number

Figure 5a shows XRD pattern of the acid-leaching products of sample PYJ at different cycle numbers. The results show that when the range of cycle number is 1–3, there is only the HH phase. When the number of cycles exceeds three, the phase peak of DH appears. According to the XRD semi-quantitative analysis, when the range of cycle number is 1–3, the HH of the acid-leaching products of sample PYJ is close to 100%. The DH and HH of the acid-leaching products of sample PYJ are 11.3% and 88.7%, respectively, at the fourth cycle. The DH of the acid-leaching products of sample PYJ is close to 100% at the fourth cycle. The reason for the phase change is that with the loss of DH crystal water, the lost crystallization water will dilute the solution. Figure 5b,c shows SEM photos of the acid-leaching products of PYJ at the second cycle and the third cycle, respectively. It can be seen that the morphology of the fibrous whiskers is almost uniform in size, with a smooth surface, clear edges and corners, and a maximum aspect ratio of 169 and 231 at the second cycle and the third cycle, respectively. At the second cycle, the length ranges from 65 μm to 193 μm, and the diameter ranges from 0.72 μm to 1.25 μm, shown in Figure 5b. At the third cycle, the length ranges from 106 μm to 231 μm, and the diameter ranges from 0.43 μm to 1.35 μm, shown in Figure 5c. Compared with Figure 4d and Figure 5b, it can be seen that with the increase of the number of cycles, the whisker grows thinner and longer.

#### 3.2.4. Effect of Reaction Time on the Crystal Habit of HH

The model of HH crystal habit regulated by the reaction time is shown in Figure 6a. The formation of HH is through the dissolution of DH and then the supersaturated nucleation process, as shown in chemical reactions (10)–(13). It can be clearly seen that the shape of the HH crystals presents a typical whisker. Therefore, the growth habit of HH is a long club shape or needle shape in the absence of a crystal modifier. According to this theory, it is better to explain that the first elongated whisker is formed, and then, the whisker becomes wider and longer with the increase of time. A part of the structure of CaSO_4_·0.5H_2_O in shown in Figure 6b. The SO_4_ tetrahedra (yellow) form, together with calcium atoms, alternate columns that arrange in a -Ca-S-Ca-S- pattern [36,37]. In the c-axis direction, two complete bonds between Ca^2+^ and SO_4_^2−^ increase the bonding stability, and the two free ends can form bonds, which results in the fastest growth along the c-axis. In the a-axis direction, because the distance between Ca^2+^ and SO_4_^2−^ is the smallest, the maximum potential energy promotes growth in the a-axis direction. Oppositely, a larger distance between Ca^2+^ and SO_4_^2−^ as well as a lower potential energy retards the growth along the b-axis [38]. The view perpendicular to the channels of the c-axis structure is shown in Figure 6c. The crystal water orders itself in the free channels in between, where one water molecule is attached to every two calcium sulfate molecules. The oxygen atoms of the water molecules point to calcium atoms, and the hydrogen atoms form hydrogen bonds to the sulfate ions [39].

### 3.3. Analysis of PYJB Sample

Figure 7a shows XRD pattern of PYJ-N6-T80-S30-X1, named PYJB, where only one phase of PYJB is HH. The main characteristic diffraction peaks of PYJB are *d*_200_ = 2.996 Å, *d*_100_ = 5.973 Å, and *d*_110_ = 3.458 Å, and cell parameters of PYJB are a = 6.937, b = 6.937, c = 6.348, α = 90 °C, β = 90 °C, and γ = 120 °C, and cell volume is 264.4 Å^3^. The sample is not much different from the cell parameters of the standard PDF card (81–1848). It can be determined that PYJB has D_3_ symmetry type, and its space group is expressed as $D_{3}^{4}-P3_{1}21$, Z = 3. Additionally, PYJB is a hexagonal system with sharp phase peaks, indicating good crystallinity. Figure 7b shows TG-DTA pattern of PYJB, in which it produces a significant thermal effect change during heating. From room temperature to 200 °C, PYJB has a narrow and large endothermic valley at approximately 164.39 °C, which is caused by the loss of crystal water of calcium sulfate (CaSO_4_·xH_2_O). Furthermore, according to the TG curve, the weight loss rate of PYJB is 6.53%. Above 200 °C, TG curve of PYJB remains unchanged. In the case of DTA curve, PYJB has an exothermic peak at approximately 217.55 °C, showing that PYJB is α-CaSO_4_·0.5H_2_O [40]. According to the relationship between the crystal water of calcium sulfate in the acid-leaching product of industrial by-product gypsum acid-leaching sample and the chemical formula (CaSO_4_·nH_2_O), the relationship between the crystal water of calcium sulfate and the chemical formula is as follows in Formula (14). This method was used to calculate the number of crystal water molecules of K_2_[Al(B_5_O_10_)]·4H_2_O [41]:G = 18*n*/((40 + 96) + 18*n*)(14)
where *n* is the number of crystal water, and G is the weight loss rate. Hence, when the calculated crystal water of PYJB is 0.53, the chemical formula of PYJB is then α-CaSO_4_·0.53H_2_O. This indicates that PYJB is similar to HH calcium sulfate, and the analysis result is consistent with those of XRD.

## 4. Conclusions

We successfully produced high-purity α-HH from PG at atmospheric pressure and low temperature by recycling reaction filtrate. Optimized reaction conditions were as follows: 6.0 mol/L HCl concentrations, 80 °C reaction temperature, and 30 min reaction time. The particle size of α-HH whiskers were uniform, and the surface was bright and smooth, leveled off, clear, and with distinct edges and corners; the maximum length to diameter ratio was 231. The cycle number was three, the length range of α-HH whiskers was from 106 mm to 231 mm, and the diameter range was from 0.43 mm to 1.35 mm, reaching the nanometer scale. It can be used as nanometer calcium sulfate hemihydrate whisker material. Analysis of whiteness was 98.6%. α-HH content reached 100% according to XRD semi-quantitative analysis, and the determined chemical formula was α-CaSO_4_·0.53H_2_O, reaching the industry double 98 standard (whiteness and purity). Therefore, PG utilization may have potential application prospect and, for the preparation of high-purity α-HH whiskers, offers a new reference. There was no solid or water waste discharge during the whole preparation process. This acid-leaching strategy could have interesting potential in producing α-HH whiskers and DH whiskers comprehensively utilizing PG. According to the prediction of 1000 YUAN /1 ton hydrochloric acid, 290.88 kg HH whiskers can be produced. The price of hemihydrate calcium sulfate on the market is CNY 6–10 per kilogram. It is estimated that the profit of treating one ton of phosphogypsum is about CNY 1900. It shows that the scheme has certain feasibility in industry. It provides new thinking for the sustainable development of the wet phosphoric acid (WPA) process industry.

## Figures and Tables

**Figure 1 materials-15-04028-f001:**
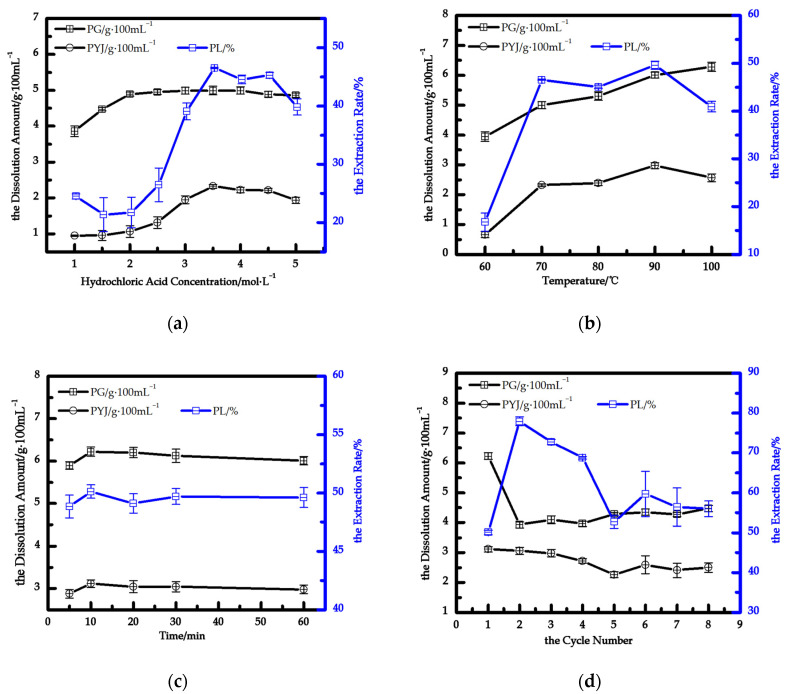
Dissolution amount of PG and crystallization amount and extraction rate of DH under different conditions (**a**–**d**): (**a**) HCl concentration; (**b**) temperature; (**c**) dissolution time; (**d**) cycle number.

**Figure 2 materials-15-04028-f002:**
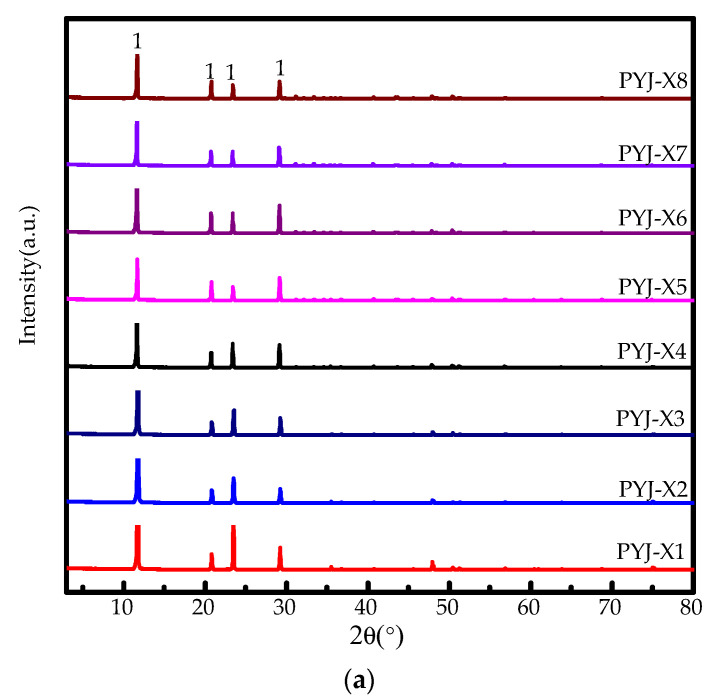
(**a**) XRD patterns (1. DH) and (**b**–**i**) SEM photos of acid-leaching products of samples PYJ at different dissolution cycles. (**b**)-1, (**c**)-2, (**d**)-3, (**e**)-4, (**f**)-5, (**g**)-6, (**h**)-7, (**i**)-8.

**Figure 3 materials-15-04028-f003:**
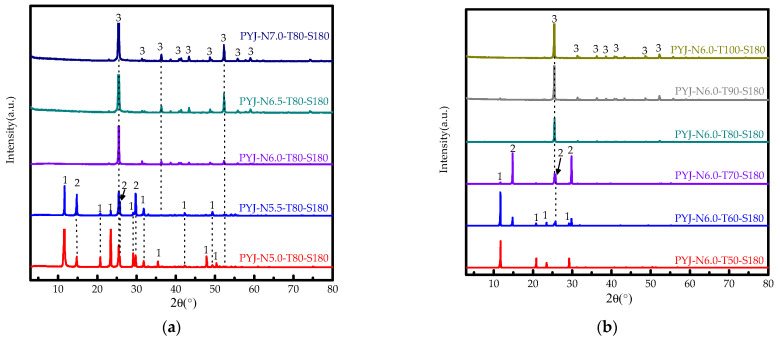
(**a**) XRD patterns in different HCl concentrations and (**b**) XRD patterns at different reaction temperatures. (1. DH, 2. HH, 3. AH).

**Figure 4 materials-15-04028-f004:**
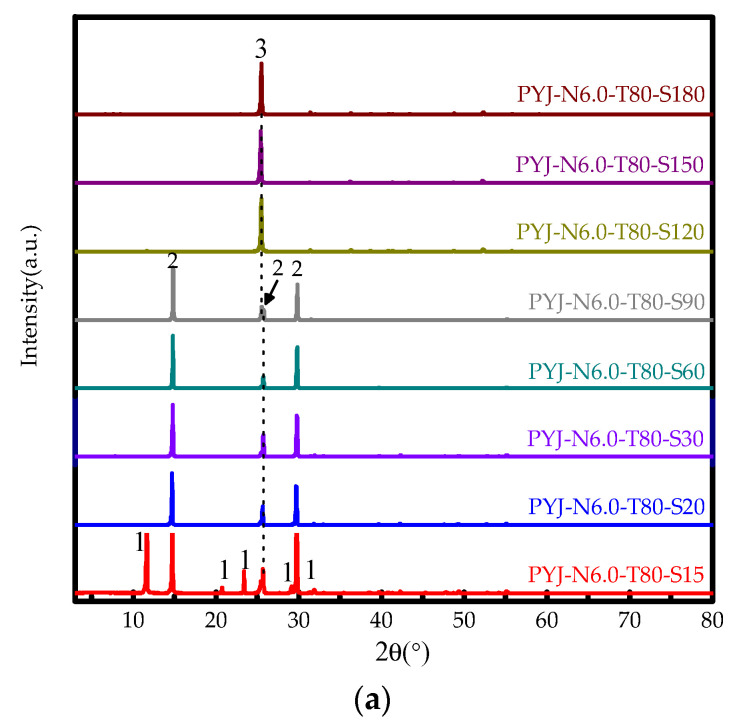
(**a**) XRD patterns (1. DH, 2. HH, 3. AH) and (**b**–**e**) SEM photos of the acid-leaching products of PYJ at different reaction times: (**b**) 15 min, (**c**) 20 min, (**d**) 30 min, (**e**) 60 min.

**Figure 5 materials-15-04028-f005:**
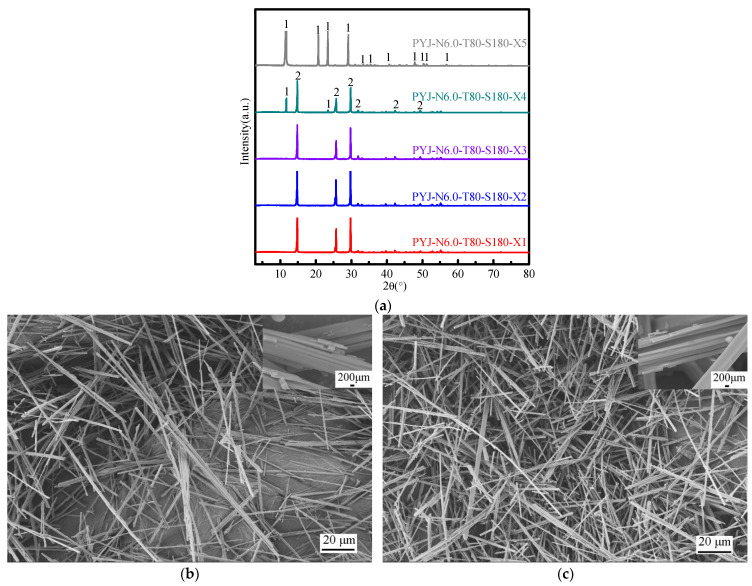
(**a**) XRD patterns (1. DH, 2. HH), and (**b**,**c**) SEM photos of the acid-leaching products of PYJ at different cycle number: (**b**) 2, (**c**) 3.

**Figure 6 materials-15-04028-f006:**
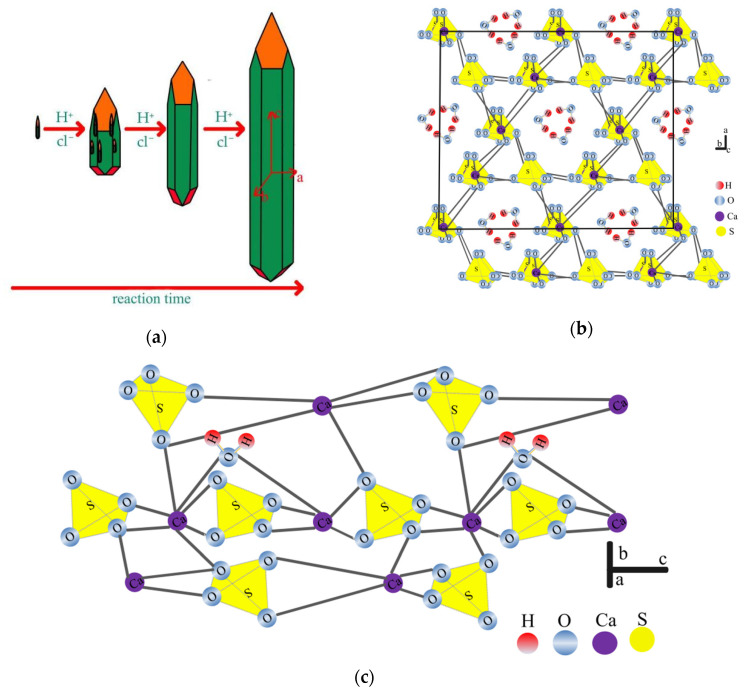
The model of CaSO_4_·0.5H_2_O crystal. (**a**) HH crystal macroscopic habit regulated by the reaction time, (**b**) A part of the microcosmic structure of CaSO_4_·0.5H_2_O, (**c**) The view erpendicular to the channels of the c-axis structure.

**Figure 7 materials-15-04028-f007:**
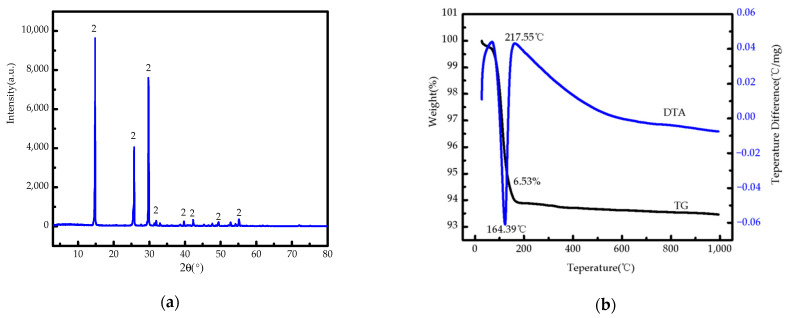
(**a**) XRD pattern (2. HH) and (**b**) TG-DTA patterns of PYJ-N6-T80-S30.

**Table 1 materials-15-04028-t001:** Chemical Composition of PG Samples by XRF (wt %).

Chemical Composition	SO_3_	CaO	SiO_2_	Al_2_O_3_	P_2_O_5_	Fe_2_O_3_	SrO	F	K_2_O	TiO_2_	MgO	BaO	Other	Loss on Ignition
wt %	39.36	29.60	5.03	1.85	1.60	0.86	0.24	0.20	0.17	0.13	0.11	0.05	0.06	20.74

**Table 2 materials-15-04028-t002:** The Radioactive Test Results of PG Samples.

Test Items		GB6656-2010					Test Results
Radioactivity		A	B		C		
	Internal Exposure Index (I_Ra_)	≤1	≤1.3	Does not satisfy class A	-	Does not satisfy class A or B	0.3
	External Exposure Index(I_γ_)	≤1.3	≤1.9		≤2.8		0.4

**Table 3 materials-15-04028-t003:** Experimental Parameters of Extraction Experiments from PG.

No.	HCl Concentration (mol/L)	Temperature (°C)	Time (min)	Cycles
1	1.0	70	60	1
2	1.5	70	60	1
3	2.0	70	60	1
4	2.5	70	60	1
5	3.0	70	60	1
6	3.5	70	60	1
7	4.0	70	60	1
8	4.5	70	60	1
9	5.0	70	60	1
10	3.5	60	60	1
11	3.5	70	60	1
12	3.5	80	60	1
13	3.5	90	60	1
14	3.5	100	60	1
15	3.5	90	5	1
16	3.5	90	10	1
17	3.5	90	20	1
18	3.5	90	30	1
19	3.5	90	60	1
20	3.5	90	20	1
21	3.5	90	20	2
22	3.5	90	20	3
23	3.5	90	20	4
24	3.5	90	20	5
25	3.5	90	20	6
26	3.5	90	20	7
27	3.5	90	20	8

**Table 4 materials-15-04028-t004:** Experimental Parameters of α-CaSO_4_·0.5H_2_O Whiskers Experiments from PYJ.

No.	HCl Concentration (mol/L)	Temperature (°C)	Time (min)	Cycles
1	5.0	80	180	1
2	5.5	80	180	1
3	6.0	80	180	1
4	6.5	80	180	1
5	7.0	80	180	1
6	6.0	60	180	1
7	6.0	70	180	1
8	6.0	80	180	1
9	6.0	90	180	1
10	6.0	100	180	1
11	6.0	80	10	1
12	6.0	80	20	1
13	6.0	80	30	1
14	6.0	80	60	1
15	6.0	80	90	1
16	6.0	80	120	1
17	6.0	80	150	1
18	6.0	80	180	1
19	6.0	80	30	1
20	6.0	80	30	2
21	6.0	80	30	3
22	6.0	80	30	4
23	6.0	80	30	5

## Data Availability

Not applicable.

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
