# Peer review of "A Simple and Efficient Method for Preparing High-Purity α-CaSO4·0.5H2O Whiskers with Phosphogypsum"

_materials, 2022, doi:10.3390/ma15114028_

Round 1

Reviewer 1 Report

The authors reported the manuscript entitled "A Simple and Efficient Method for Preparing high-purity α -CaSO4·0.5H2O whiskers with phosphogypsum" very well.  Hydrated calcium sulphate has extracted in order to purify Phosphogypsum. Manuscript is well written and nicely organised. 

Paper is useful to the scientific society but some points should be cleared before publication.

  1. The novelty of work should be included.
  2. Why do the dissolution amount of PG increase and the impurity of PG easily leach with increasing temperature?
  3. When the temperature in the solution increases, gamma decreases, and concentration c increases. When S>1, crystals form naturally from oversaturated solutions [30]. Please check the sentence It looks likes incomplete...
  4. Why does the temperature in the solution increase, gamma decrease, and concentration c increase?
  5. Please insert a clear figure of XRD.
  6. The peak intensity of DH decreases first and then disappears, and the peak intensity of HH  increases first and finally disappears, whereas that of AH phase trends from zero and gradually increases with the increase of reaction time. Why? Explain properly.
  7. English language should be checked properly.

  8. What is the concentration of HCl for leaching?
  9. Why the crystallizing amount of DH gradually increases and then decreases with increasing HCl concentration?
  10. Background of first page figure should be changed for better visibility.

Reviewer 2 Report

A method for preparing high-purity phosphogypsum - a waste material during mineral fertilizer production is presented. The topic is timely and of interest.
Here some more suggestions to further improve the quality of this work:
-I suggest focussing on the high level results of your research raterh than results from specific experiments. Why is this important? Is this applicable to waste phosphogypsum in China or elsewhere or is it rather a laboratory study that is not applicable to industrial application. I am from industry and for us it is always important to see if these results can actually be used in large scale applications. Is this the case here?
-we usually call it wet phosphoric acid (WPA) process
-I am not aware of wide use of PG. I feel these are largely scientific works while the PG cannot be properly utilized on industrial scale, hence the accumulation of 200 million tons per year that you also acknowledge.
-Isn't there also a law in China that PG should not be stockpiled in China anymore? If so this should be mentioned here.
-In the US PG is a huge problem since it is a big liability for the tax payer and Europe seems to be looking at ways to use PG as a gypsum replacement (https://doi.org/10.1016/j.resconrec.2022.106328), this can also be mentioned here
-you mention that it is a high value material could you provide prize estimates - and the cost of the PG - I think this would be helpful
-I think the radioactivity of the material would be really interesting and also the source of the PG and the phosphate rock that was used as a raw material
-HCl is relatively expensive, I think a quick cost benefit analysis would be very helpful
-I don't find the XRD patterns particularly helpful
-I am not sure I can follow what you did here - I find it hard to understand the train of thought, in any case I feel it is paramount that radiation measurements are conducted to see if this material could actually be used in industry.

Round 2

Reviewer 1 Report

The authors have revised the manuscript in light of the referee's comments nicely. The Paper may be accepted for publication in its current form.

Reviewer 2 Report

Thank you for looking through this one more time. The manuscript is fine from my end and I am excited to see it getting published.